# Feeding System Resizes the Effects of *DGAT1* Polymorphism on Milk Traits and Fatty Acids Composition in Modicana Cows

**DOI:** 10.3390/ani11061616

**Published:** 2021-05-29

**Authors:** Serena Tumino, Andrea Criscione, Valentina Moltisanti, Donata Marletta, Salvatore Bordonaro, Marcella Avondo, Bernardo Valenti

**Affiliations:** 1Department of Agriculture, Food and Environment, University of Catania, Via Valdisavoia 5, 95123 Catania, Italy; serena.tumino@unict.it (S.T.); acriscio@unict.it (A.C.); valentina.moltisanti@gmail.com (V.M.); d.marletta@unict.it (D.M.); s.bordonaro@unict.it (S.B.); 2Department of Agricultural, Food and Environmental Science, University of Perugia, Borgo XX Giugno 74, 06121 Perugia, Italy; bernardo.valenti@unipg.it

**Keywords:** *DGAT1* K232A polymorphism, Modicana cows, pasture feeding, milk traits, milk fatty acids

## Abstract

**Simple Summary:**

Genetic selection for single-locus polymorphisms could offer suitable opportunities to rapidly improve milk traits in local unselected cattle breeds characterized by low production levels. Since these hardy breeds are generally raised in traditional extensive and semi-intensive systems, which make wide use of grazing resources, the interactive effect between genotype and feeding system is worthy of investigation. In Modicana cattle breed, milk composition and fatty acid profile were influenced by both genetic polymorphisms at the *DGAT1* K232A locus and feeding systems. The milk from homozygous AA cows was associated with a more favorable fatty acid composition due to a lower percentage of total saturated fatty acids, saturated to unsaturated ratio, atherogenic index, and a greater presence of oleic acid and total unsaturated fatty acids. Our finding confirmed the important role of pasture feeding on milk composition: the high nutritional and healthy value of milk obtained in extensive systems by pasture-fed cows. The interaction between the two experimental factors also appears to play a role: in our experimental condition, it seems that high pasture feeding can resize the effect of the *DGAT1* genotype on milk traits and fatty acid composition in Modicana cows.

**Abstract:**

The interaction between genetic polymorphism and feeding system on milk traits and fatty acid composition was investigated in Modicana cows. Two *DGAT1* K232A genotypes (AK and AA) and two feeding regimes, extensive system (EX) with 8 h of grazing without concentrate (EX) and semi-intensive systems (SI) with 2 h of grazing with concentrate, were investigated. *DGAT1* genotype did not influence milk yield and composition. The feeding system affected milk composition: protein was significantly higher in SI and lactose in the EX system. A significant genotype × feeding system interaction was observed: the protein and casein levels of AK cows were higher in the SI compared to the EX system. Milk fatty acids profile, total saturated to total unsaturated fatty acids, n-6 to n-3 ratios, and atherogenic index were affected by the feeding system, improving the healthy properties of milk from animals reared in the extensive system. *DGAT1* genotype influenced the fatty acid composition: milk from AA cows had a more favorable fatty acid composition due to lower total saturated fatty acids, saturated to unsaturated ratio, atherogenic index, and higher levels of oleic acid and total unsaturated fatty acids. Furthermore, an interaction genotype x feeding system was observed: the AK milk was richer in short-chain FAs (C4:0–C8:0) and C10:0 only in the EX but not in the SI system. Our data suggest that a high amount of green forage in the diet of Modicana cows can resize the effect of the *DGAT1* genotype on milk traits and fatty acids composition.

## 1. Introduction

In cattle, feeding systems greatly affect milk quality traits such as chemical and fatty acid composition. In particular, it is widely accepted that a pasture-based diet increases the levels of polyunsaturated fatty acids, isomers of conjugated linoleic acid (CLA), and omega-3 fatty acids [1].

Bovine milk fatty acids composition is a complex trait affected by genetic and environmental factors, such as breed, diet, and stage of lactation [2,3,4]. More recently, the interaction between genetic polymorphism and the feeding system has also been reported [5,6].

Candidate gene and quantitative trait locus (QTL) mapping approaches have been implemented to detect genes or QTLs for milk fatty acid traits. One of the major QTL affecting milk fat composition in dairy cattle is located in the centromeric region of BTA14 and includes the *DGAT1* gene that encodes for the diacylglycerol acyltransferase1 [7]. This enzyme catalyzes the final step of triglyceride synthesis by the esterification of fatty acyl-CoA to the sn-3 position of diacylglycerol [8]. The *DGAT1* coding sequence is 8.6 kb long and consists of 17 exons, but the most widely studied and validated polymorphism is located in the exon VIII [9]. This polymorphism consists in a substitution of the lysine (K) with alanine (A) in position 232 (K232A) of the amino acid sequence, caused by the non-conservative dinucleotide substitution (AA > GC) of adenine and adenine (A-A) by guanine and cytosine (G-C), resulting in a change of enzyme functional properties. In fact, in vitro evidence revealed that the two protein isoforms of *DGAT1* differed in their ability to synthesize triglycerides [10]. The effects of the *DGAT1* K232A polymorphism on milk production traits have been widely investigated in different cattle breeds. In particular, the K variant has been associated with increased milk fat percentage and saturated fatty acids [11,12,13,14,15,16,17].

Local breeds are fully adapted to traditional extensive production systems, which are characterized by low inputs, capital, and labor and by a lower productivity per animal and per surface compared to farmed land areas. Dairy products derived from pasture-based systems, still present in many marginal areas, are perceived as natural and healthy by the consumers. The rustic breeds represent significant genetic resources to satisfy present and future demands for sustainable farming in a changing environment [18]. In Sicily, the Modicana cattle breed is traditionally reared in extensive and semi-intensive systems; in the province of Ragusa, its milk is mainly used to produce “Ragusano”, a protected designation of origin (PDO) pasta-filata cheese. The feeding system of Modicana cows is mainly extensive, essentially based on the exploitation of the natural pasture with no or limited supplementation with concentrate; a semi-intensive feeding system is also present in the breeding area, where the pasture feeding is limited to few hours per day and a regular concentrate supplementation is always present.

Despite its historical relevance and economic importance in the local rural community, Modicana has been poorly investigated for candidate genes, and only some major genes [5,6,19,20] were investigated in this dairy breed.

This research investigates the effect of the genetic polymorphism at the DGAT1 K232A locus and its interaction with the feeding system on milk productions and composition in the Modicana cattle breed. Our hypothesis is that a different pasture-feeding rate could modify the effect of *DGAT1* polymorphism on milk traits and fatty acid profile.

## 2. Materials and Methods

### 2.1. Animals and Experimental Design

Forty multiparous (3rd–4th lactation) Modicana cows were selected from two Sicilian farms characterized by two feeding systems: an extensive (EX) and a semi-intensive (SI) system.

In the EX farm, cows were fed with mixed hay and natural pasture (8 h per day). The SI farm diet was based on mixed hay, concentrate (maize meal 30%, barley meal 32%, beet pulp 12%, wheat bran 8%, soybean meal 11%, carob pulp 4%, mineral, and vitamin mix 3%), and natural pasture (2 h per day). The botanical composition of pasture in both farms was dominated by different species of compositae, cruciferae, and malvacee, and, to a much lesser extent, poaceae and fabaceae.

In each farm, the animals were chosen based on their genotype at the *DGAT1* K232A locus. The cows, homogeneous for the stage of lactation and milk yield (EX: 94 ± 25.2 days of lactation and 14.3 ± 4.7 kg of milk yield per day; SI: 105 ± 38.7 days of lactation and 15.0 ± 3.8 kg of milk yield per day) were distributed as follows:EX: AA, 15 cows; AK, 5 cows;SI: AA, 15 cows; AK, 5 cows.

### 2.2. Genetic Characterization

The polymorphism AA > GC at the VIII exon of the *DGAT1* locus (Acc. Num. AY065621) was analyzed by PCR-RFLP according to Komisarek et al. [21]. The dinucleotide substitution leading to a single amino acid change in the translated protein sequence was identified as K232A; genotypes were indicated as AA and AK. Genomic DNA was isolated from milk samples collected during daily milking operations by using a commercial kit (the Norgen Biotek Milk DNA Preservation and Isolation Kit, Thorold, ONT, CA). The target DNA sequence was amplified starting from 50 ng/µL of DNA in a total reaction mixture of 30 µL using a GenAmp PCR System 9700 (Applied Biosystems, Foster City, CA, USA) thermal cycler. The thermal amplification profile consisted of initial denaturation step at 94 °C for 5 min followed by 30 cycles of denaturation at 94 °C for 30 s, annealing at 58.5 °C for 30 s and extension at 72 °C for 40 s, and a final extension of 72 °C for 5 min. The amplicons were subjected to digestion for 2 h with 5 units of BglI restriction enzyme (New England BioLabs Inc., Ipswich, MA, USA). Digested fragments were visualized on 3% agarose gel stained with GelRed Nucleic Acid Stain (Biotium, Inc., Fremont, CA, USA). Restriction pattern was electronically recorded using the ChemiDocTM System with Image LabTM Software (Bio-Rad Laboratories, Inc., Segrate, Italy).

The 378 bp amplified region containing the exon VIII of *DGAT1* gene was sequenced on both strands in 6 samples (3 for each genotype AA and AK) on an ABI PRISM 3130 Genetic Analyzer (Applied Biosystems, Foster City, CA, USA). The obtained sequences were aligned with the reference sequence (GenBank Acc. Num. M57764.1) by adopting the software CLUSTALX 1.8 [22] in order to identify potential novel SNPs.

### 2.3. Data Collection and Chemical Analysis

The trial lasted 8 months, from December to July. Monthly, milk yield was individually measured, and samples were collected from morning and afternoon milkings. Milk samples, consisting of proportional volumes of the two daily milkings, were analyzed for fat, protein, lactose, casein, and urea using an automated Fourier transform infrared absorption spectrophotometric analyzer (Combi-foss 6000, Foss Electric, Hillerød, Denmark).

Body condition score was estimated individually at the start and at the end of the trial. Milk fat was extracted and converted to fatty acid methyl esters, according to Campione et al. [23]. Specifically, milk samples were centrifuged at 17,800× *g* for 30 min at 4 °C. The upper-fat layer cake was transferred to a microtube, left at room temperature for 30 min before, and centrifuged at 19,300× *g* for 20 min at room temperature. Eventually, the top layer was collected for fatty acid methyl esters (FAME) preparation. The FAME were obtained by base-catalyzed transesterification procedure (Christie, 1982) using sodium methoxide in methanol 0.5 N and nonadecanoic acid (Sigma Chemical Co., St. Louis, MO, USA) as an internal standard. Milk fatty acid composition was analyzed as reported by Valenti et al. [5]. Individual FAME were separated using a Trace Thermo Finnigan gas-chromatograph equipped with a flame ionization detector (FID; ThermoQuest. Milan, Italy) and a 100 m high-polar fused silica capillary column (25 mm i.d., 0.25 μm film thickness; SP. 24056; Supelco Inc., Bellefonte, PA, USA). Helium was the carrier gas at a constant flow of 1 mL/min. Total FAME profile in 1-μL-sample volume at a split ratio of 1:80 was determined using the following gas-chromatographic conditions: the oven temperature was set at 50 °C and held for 4 min, then increased to 120 °C at 10 °C/min, held for 1 min, then increased up to 180 °C at 5 °C/min, held for 18 min, then increased up to 200 °C at 2 °C/min, held for 15 min, and then increased up to 230 °C at 2 °C/min, held for 19 min. The injector and detector temperatures were at 270 °C and 300 °C, respectively. FAME identification was based on a commercial mixture of standard FAME (Nu-Chek Prep Inc., Elysian, MN, USA) and individual standard FAMEs (Larodan Fine Chemicals, Malmo, Sweden) and expressed as g/100 g of total FAME.

### 2.4. Statistical Analysis

The Hardy–Weinberg equilibrium at the DGAT1 locus was verified using the chi-squared test. Individual data for milk yield and composition (fat, protein, lactose, fatty acids) were investigated using the GLM procedure for repeated measures of Statistical Package for Social Science (SPSS for Windows, SPSS Inc., Chicago, IL, USA). The analysis included the main effects of *DGAT1* genotype (AA, AK), feeding system (EX, SI), month of milk collection, and the interaction genotype × feeding system. Pre-experimental data of milk production were used as a covariate for milk yield and composition. The covariate was excluded from the analysis if it was not significant (*p* > 0.05). When the interaction showed significant values, differences between means were evaluated using Tukey’s adjustment for multiple comparisons. Significance was declared when *p* ≤ 0.05.

## 3. Results

Sequence analysis confirmed the observed polymorphism, and no further variant was found in our samples in the analyzed amplicon of the *DGAT1* gene. The effects of *DGAT1* genotype and feeding system on milk yield and gross composition in Modicana cows were reported in Table 1. No significant difference in milk yield and composition was associated with the *DGAT1* AK and AA genotype. The feeding system affected the percentage of milk protein, which was significantly higher in the SI farm, and lactose, significantly higher in the EX farm. A genotype × feeding system interaction was observed for the percentage of protein and casein, both significantly higher in the heterozygous AK cows reared in the SI as compared to heterozygous AK cows reared in the EX system (Figure 1). Indeed, the casein level was significantly higher in AK than in AA cows only in the SI system. BCS was not affected by genotype and feeding system (data not reported).

Detailed milk fatty acid (FA) composition is reported in Table 2. *DGAT1* AK genotype was significantly associated with higher levels of C8:0, C11:0, total SFA, SFA/UFA ratio and atherogenic index (AI), and to lower levels of C18:1*c*9, total desaturation index, total unsaturated fatty acid (UFA), and total unsaturated C18, in comparison with AA genotype.

The feeding system affected the milk fatty acid profile. Specifically, all short-chain fatty acids (C4:0–C8:0), C17:0, C18:0, almost all unsaturated FA, total polyunsaturated FA (PUFA), total trans FA, total UFA, and total n-3 FA were higher in the EX system, whereas C:11, C12:0, C14:0, C16:0, C15:0 *iso*, C17:0 *iso*, 17:0 *anteiso*, the monounsaturated FA C12:1, C14:1, C16:1, and n-6/n-3 ratio were higher in the SI system. Desaturation indices (C14:1, C16:1, C18:1, and CLA ratios) were higher in the SI system.

A significant interaction between *DGAT1* genotype and feeding system was found for C4:0, C6:0, C8:0, C10:0 (Figure 2): in particular, within the EX system, the AA cows showed lower values, compared to AK cows. On the contrary, within the SI system, no difference in the above-mentioned fatty acids was evident between genotypes.

## 4. Discussion

The genetic background of local breeds is still scarcely investigated in comparison to that of cosmopolitan ones, despite the availability of increasingly advanced tools for genomic investigation. In these breeds with low production levels, genetic variation of milk traits associated with a single-locus polymorphism may offer the opportunity for changing milk yield and composition by selecting individuals based on their genotype. A low frequency of the *DGAT1* K allele (0.09) in 165 Modicana cows has been reported by Valenti et al. [5], in agreement with the values observed in several Italian dairy and dual-purpose breeds [12,27]. Recently Fink et al. [28], using a large mammary RNA seq dataset, in conjunction with in vitro expression experiments, have found that animals carrying K allele showed greater mammary *DGAT1* expression compared to those with A allele, suggesting that most of the milk traits variations due to *DGAT1* polymorphism may derive from a differential expression based mechanism.

In our experimental conditions, BCS (data not reported), milk yield, and gross composition were not affected by the K232A *DGAT1* genotype. The effect of this polymorphism on milk production is quite controversial. Most studies report higher milk yield in cows carrying the A allele than the K allele [15,29,30], but no significant differences are reported [31,32]. In contrast, it is well recognized that cows carrying K allele produce milk with higher fat concentration compared to cows carrying the A allele [14,15,17,30]. It has been reported that the effect size of *DGAT1* p.K232A polymorphism on fat concentration could differ among breeds, probably due to existing interactions with genetic background in the different populations [29,33,34,35]. To our knowledge, no results are available on the role of *DGAT1* genotype on milk traits in the Modicana breed, but it is possible to hypothesize that the lack of effect could be due to the breed used in this study.

The feeding system influenced the milk protein, higher in SI system and lactose, higher in EX system. A significant interaction genotype x feeding system was evident for protein and casein levels. The two genotypes showed a different response to the feeding system: the protein and casein levels of AK cows were higher in the SI compared to the EX system. The same difference due to the feeding system was not evident for AA cows. Indeed, in the SI system, the AK cows showed higher casein levels compared to AA cows. It has been shown that the presence of the K allele leads to more protein synthesis compared to the A allele [12,29,30,36]. Given the role of dietary starch on milk protein synthesis [37,38], it could be hypothesized that cows carrying the K allele better exploited their potential ability to produce more protein in the presence of a starch-rich feed (i.e., concentrate) in the diet, as occurred in the SI system, compared to the EX system, where the diet was mainly based on pasture. It is supposable that the same difference was not appreciated in the AA group due to the lower genetic aptitude to produce protein.

The diacylglycerol acyltransferase1 catalyzes the esterification of fatty acyl-CoA to the sn-3 position of a diacylglycerol. First data on the influence of *DGAT1* polymorphism on milk fatty acids were reported by Schennink et al. [11], which found higher C16:0 and SFA/UFA ratio and lower C14:0, C18 unsaturated fatty acids and CLA in cows with K allele, compared to A allele. Similar results were also reached by other authors [14,15,34]. Consistently, in our conditions, the K allele was associated with higher SFA (C8:0, C11:0, and total SFA), lower C18 unsaturated FA, and higher SFA/UFA ratio. C16:0 showed a non-significant (*p* = 0.056) trend toward a higher AK genotype level compared to AA, whereas no differences were evident in C14:0 and CLA levels.

The lower SFA/UFA ratio in the AA group led to a lower atherogenic index, calculated as proposed by Ulbricht and Southgate [24]. As a consequence of the differences in the unsaturated fatty acids content, the total unsaturation index was higher in the AA group. Schennink et al. [26] found lower C10:0, C12:0, C14:0, and C16:0 and higher C18:0, CLA, and total desaturation indices in milk from cows carrying A allele, compared to K allele. Coherently, Houaga et al. [16] found that the A allele was associated with higher C18:0 and total unsaturation indices. The results relating to the aforementioned indicators seem to highlight an effect of the *DGAT1* genotype on the milk’s health properties, which, in light of the results obtained in our and other experimental conditions, seem to improve in the presence of the A allele.

Milk fat composition was greatly affected by the feeding system. Most of the fatty acids differed between EX and SI diets. In our conditions, a greater percentage of total UFA, PUFA, -trans FA, and n-3 FA and less SFA and medium-chain fatty acids were found in milk from cows reared in the EX system. The differences observed between the two feeding systems are consistent with most of the studies reporting the milk fat composition in relation to dietary fresh forage [1,39]. All desaturation indices apart from the total index, the n-6/n-3, and the AI were lower in milk from animals reared in the EX system. Short-chain fatty acids, C4:0, C6:0, and C8:0, were higher in the EX group. The higher level of pasture in the EX diet seems to justify the increased short-chain FA, as shown by Abel et al. [40] for C4:0 and C6:0 when a pasture-based diet and a total mixed ration (TMR) were compared. Other authors report no effect on milk short-chain fatty acids between pasture and TMR diets [41,42].

The interaction genotype × feeding system caused a different response of *DGAT1* polymorphism in the two feeding systems for C4:0, C6:0, C8:0, and C10:0. In the EX system, the AK cows showed a greater percentage of these fatty acids in comparison with AA cows, whereas, in the SI system, no difference was evident between the two genotypes. Significant interactions between *DGAT1* genotype × season are reported by Pacheco-Pappenheim et al. [17], according to seasonal variation of the dietary pool of fatty acid. The authors found that milk from KK cows was higher in short-chain fatty acids during summer, when fresh herbage was available, and lower during winter, compared to AA cows. Different triacylglycerol (TAG) levels were also observed between genotypes during summer but not during winter. The authors suggested that the dietary fat variations due to season could have affected the fatty acids esterified by the different *DGAT1* genotypes during TAG synthesis. Duchemin et al. [43] report that the effect of *DGAT1* A allele in determining an increase in UFA was more pronounced in summer, when fresh herbage was available than in winter, with an indoor feeding. Similar results due to season or feeding system are reported for *SCD1* [5,43]. It is conceivable that at the basis of these results, there may be a different expression of the lipogenic genes in relation to the diverse availability in the diet of crude fiber and fatty acids, responsible for the development of acetate and butyrate in the rumen and de novo fatty acids in milk.

## 5. Conclusions

This study investigated for the first time in Modicana cows the effect of the polymorphism at the DGAT1 locus and its interaction with the feeding system on milk production and composition. Rearing the AK cows in the semi-intensive feeding system increased the protein and casein levels in comparison with the extensive system, suggesting a better efficiency in the use of dietary energy from concentrate in these conditions. Our data confirmed the major role of the feeding system in improving the fatty acid composition of milk. However, the genetic polymorphism of *DGAT1* also played a role. In particular, the milk from AA cows was associated with a more favorable fatty acid composition due to a lower percentage of total SFA, SFA-to-UFA, atherogenic index, and a greater presence of oleic acid and total UFA. In addition, a different response to the feeding system was recorded between the two genotypes, with lower short-chain FA in AA cows only in the extensive system. Our data suggest that genetic selection for single-locus polymorphism could offer suitable opportunities to manipulate milk traits in the Modicana cow. Nevertheless, the genetic selection should take into account the context of the feeding system both for its direct effect on milk quality traits and for its ability to resize the genetic effects.

## Figures and Tables

**Figure 1 animals-11-01616-f001:**
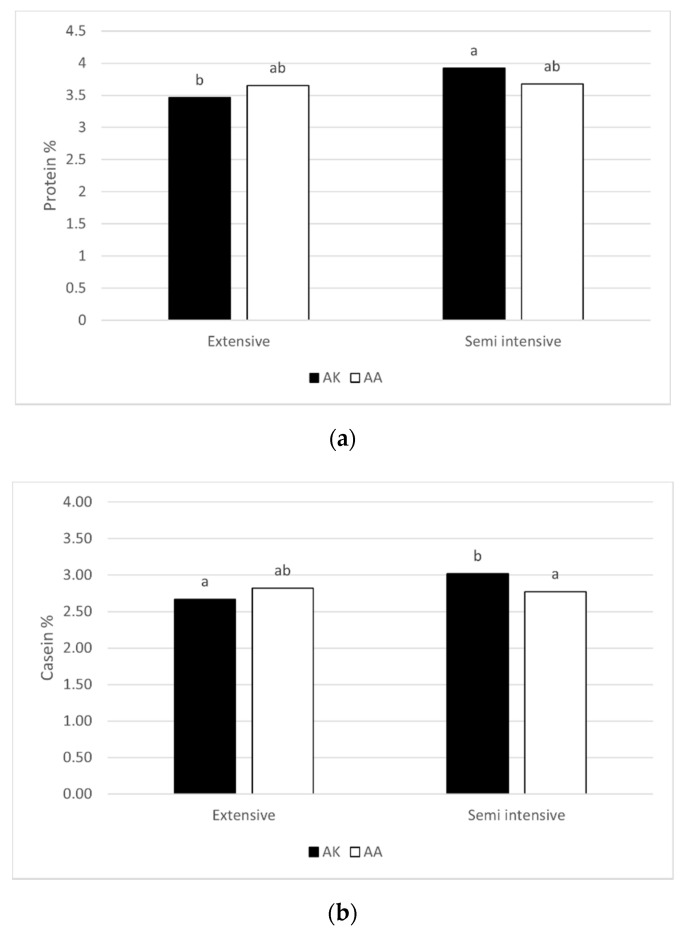
Effect of the interaction between genotype at *DGAT1* K232A (AK and AA) and feeding system (extensive and semi-intensive) on protein (**a**) and casein (**b**) percentages in Modicana cow milk; ^a,b^ Different superscript letters indicate differences (*p* ≤ 0.05) between mean values tested by Tukey’s multiple comparison test.

**Figure 2 animals-11-01616-f002:**
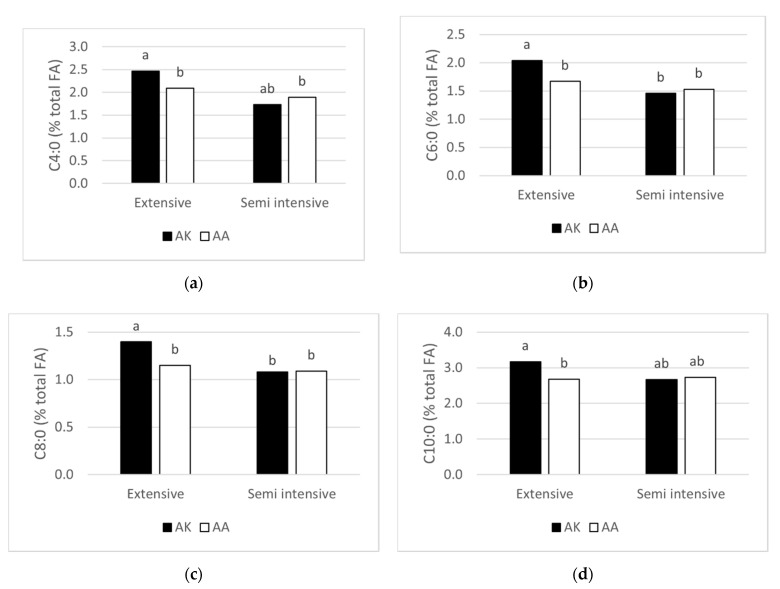
Effect of the interaction between genotype at *DGAT1* K232A (AK and AA) and feeding system (extensive and semi-intensive) on C4:0 (**a**), C6:0 (**b**), C8:0 (**c**), and C10:0 (**d**) fatty acids. ^a,b^ Different superscript letters indicate differences (*p* ≤ 0.05) between mean values tested by Tukey’s multiple comparison test.

**Table 1 animals-11-01616-t001:** Effects of *DGAT1* K232A genotype and feeding system on mean milk yield and composition in Modicana cow.

Milk Yield and Composition	Genotype (G)		Feeding System (S)		Significance			SEM
	AK	AA	EX	SI	G	S	G × S	
Milk yield kg/d	8.73	9.57	8.65	9.65	0.243	0.174	0.135	0.378
Fat %	3.86	4.03	4.01	3.87	0.387	0.489	0.542	0.070
Protein %	3.69	3.67	3.56 ^b^	3.80 ^a^	0.747	0.008	0.015	0.031
Lactose %	4.46	4.44	4.59 ^a^	4.31 ^b^	0.865	0.000	0.255	0.027
Urea mg/dL	20.6	24.5	22.6	22.5	0.213	0.980	0.598	1.020
Casein %	2.78	2.78	2.75	2.81	0.538	0.071	0.015	0.026

EX = extensive system; SI = semi-intensive system. ^a,b^ Different superscripts within a row indicate a significantly different values at *p* ≤ 0.05.

**Table 2 animals-11-01616-t002:** Effects of DGAT K232A genotype and feeding system on mean milk fatty acid composition.

Milk Fatty Acids % of Total Fatty Acids	Genotype (G)		Feeding System (S)		Significance			SEM
	AK	AA	EX	SI	G	S	G × S	
C4:0	2.10	1.99	2.28 ^a^	1.81 ^b^	0.328	0.000	0.019	0.06
C6:0	1.75	1.60	1.86 ^a^	1.50 ^b^	0.070	0.000	0.012	0.04
C8:0	1.24 ^a^	1.12^b^	1.28 ^a^	1.08 ^b^	0.040	0.001	0.026	0.03
C10:0	2.92	2.70	2.92	2.70	0.097	0.096	0.042	0.06
C11:0	0.35 ^a^	0.30^b^	0.29 ^a^	0.37 ^b^	0.018	0.000	0.897	0.01
C12:0	3.51	3.26	3.23 ^b^	3.53 ^a^	0.093	0.047	0.265	0.07
C12:1	0.21	0.20	0.16 ^b^	0.25 ^a^	0.229	0.000	0.665	0.01
C14:0	11.28	11.02	10.44 ^b^	11.86 ^a^	0.471	0.000	0.395	0.19
C15:0 *iso*	0.42	0.40	0.37 ^b^	0.46 ^a^	0.373	0.000	0.845	0.01
C15:*0 anteiso*	0.79	0.81	0.77	0.82	0.635	0.129	0.907	0.01
C14:1 *c*9	1.05	0.89	0.69 ^b^	1.26 ^a^	0.066	0.000	0.379	0.06
C15:0	1.59	1.59	1.58	1.61	0.988	0.477	0.894	0.02
C16:0	30.22	28.21	26.97 ^b^	31.45 ^a^	0.056	0.000	0.904	0.58
C17:0 *iso*	0.46	0.48	0.44 ^b^	0.49 ^a^	0.167	0.009	0.436	0.01
C17:0 *anteiso*	0.54	0.60	0.45 ^b^	0.69 ^a^	0.075	0.000	0.264	0.02
C16:1 *c*9	1.49	1.40	1.20 ^b^	1.68 ^a^	0.309	0.000	0.887	0.05
C17:0	0.72	0.77	0.78^a^	0.70 ^b^	0.084	0.005	0.111	0.02
C18:0	8.89	9.36	10.20^a^	8.05 ^b^	0.254	0.000	0.863	0.24
C18:1 *t*11	1.42	1.50	2.16^a^	0.77 ^b^	0.490	0.000	0.488	0.13
C18:1 *c*6	0.60	0.73	0.84^a^	0.49 ^b^	0.186	0.001	0.393	0.05
C18:1 *c*9	19.67 ^b^	21.21 ^a^	20.26	20.62	0.048	0.632	0.916	0.33
C18:1 *c*11	0.48	0.51	0.51	0.48	0.400	0.343	0.832	0.01
C18:2	1.61	1.65	1.65	1.61	0.553	0.545	0.671	0.02
C18:3 *alfa*	0.78	0.84	1.13 ^a^	0.49 ^b^	0.226	0.000	0.248	0.06
CLA *c*9*t*11	0.66	0.71	0.92 ^a^	0.45 ^b^	0.337	0.000	0.104	0.05
n-3	0.96	1.02	1.44a	0.55b	0.310	0.000	0.248	0.78
n-6	1.89	1.99	1.95	1.93	0.387	0.817	0.203	0.46
n-6/n-3	2.55	2.50	1.36b	3.68a	0.751	0.000	0.608	0.20
SFA	62.28 ^a^	59.66 ^b^	59.54 ^b^	62.39 ^a^	0.025	0.016	0.186	0.59
MUFA	23.51	24.95	23.66	24.79	0.057	0.131	0.731	0.33
PUFA	3.51	3.71	4.31 ^a^	2.92 ^b^	0.223	0.000	0.084	0.14
Trans	1.81	1.92	2.62 ^a^	1.11 ^b^	0.414	0.000	0.472	0.14
UFA	28.94 ^b^	30.69 ^a^	30.74 ^a^	28.89 ^b^	0.046	0.035	0.457	0.41
SFA/UFA	2.18 ^a^	1.97 ^b^	1.96 ^b^	2.18 ^a^	0.028	0.023	0.280	0.05
AI	6.34 ^a^	5.65 ^b^	5.54 ^b^	6.44 ^a^	0.033	0.007	0.491	0.16
Desaturation indexes *								
C14 ratio *	0.08	0.07	0.06 ^b^	0.10 ^a^	0.073	0.000	0.303	0.00
C16 ratio *	0.05	0.05	0.04 ^b^	0.05 ^a^	0.924	0.002	0.862	0.00
C18 ratio *	0.69	0.69	0.66 ^b^	0.72 ^a^	0.598	0.000	0.606	0.01
CLA ratio *	0.34	0.33	0.30 ^b^	0.37 ^a^	0.518	0.000	0.151	0.01
Total Index **	0.31 ^b^	0.33 ^a^	0.32	0.32	0.046	0.874	0.607	0.00

EX = extensive system; SI = semi-intensive system; SFA: saturated fatty acids; MUFA = monounsaturated fatty acids; PUFA = polyunsaturated fatty acids; AI, atherogenic index [C12 + 4 (C14) + C16]:(sum of unsaturated FA) [24]; * Desaturation index calculated as reported by Kelsey et al., [25] is the ratio between *c*9- fatty acid and the sum of its saturated homologous plus the *c*9 fatty acids (e.g., *c*9 14:1/(14:0+*c*9 14:1)); CLA ratio (CLA *c*9*t*11/(CLA*c*9*t*11+C18:1 *t*11) ** Total Index = [(C10:1 + C12:1 + C14:1 *c*9 + C16:1 *c*9 + C18:1 *c*9 + CLA *c*9, *t*11)/(C10:1 + C12:1 + C14:1 *c*9 + C16:1 *c*9 + C18:1 *c*9 + CLA *c*9, *t*11 + C10:0 + C12:0 + C14:0 + C16:0 + C18:0 + C18:1 *t*11)] × 100 [26]. ^a,b^ Different superscripts within a row indicate significantly different values at *p* ≤ 0.05.

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
