# Peer review of "Feeding System Resizes the Effects of *DGAT1* Polymorphism on Milk Traits and Fatty Acids Composition in Modicana Cows"

_animals, 2021, doi:10.3390/ani11061616_

Round 1
Reviewer 1 Report
Please check your summary once again. Not all statements are clear, especially if you connect sayings in one sentence (talking about respectively).
Please try to shorten your conclusions further. It is still a lot of discussion in it.
Author Response
We proceeded to revise the text according to the suggestions of referee 1, as highlighted in the text attached with the revisions. Hoping that the paper is now worthy of publication on Animals, we send our best regards
Reviewer 2 Report
This resubmitted manuscript was significantly improved by authors according to my review opinion, so I accepted this manuscript.
Author Response
none
Reviewer 3 Report
Satisfied with the authors responses to my previous comments
Author Response
none
This manuscript is a resubmission of an earlier submission. The following is a list of the peer review reports and author responses from that submission.
Round 1
Reviewer 1 Report
Dear authors,
This is an interesting work, and its novelty is unquestionable as it shows that high pasture feeding can moderate the effect of DGAT1 genotype on milk production traits and fatty acids composition in Modicana cows.
The findings here is publishable, with the following minor corrections.
Comment-Line 109: Since genomic DNA was isolated from milk samples, I guess that explains why no animal ethics consideration was highlighted in your materials and methods.
Line 111: Check the “close bracket”.
Line 111-112: Since the target DNA sequence was amplified starting from 50 ng/uL of DNA in a total reaction mixture of 30uL, what was the thermal profile of the PCR reaction?
Line 197: The first sentence needs to be revised
Line 230: Consider revising “Potentially” to “Potential”.
Reviewer 2 Report
The aim of Tumino et al. manuscript is very interesting because of the importance of local breeds’ milk production and improving milk quality since the point of view of human health. However, I think that this manuscript must be improved intensively.
Title: title is incorrect, must be change! And add local cattle name!
Simple abstract: completely rewrite! This is different than abstract!!
line (l) 31: “higher in SI and EX farm,” this sentence is not correct, please correct it!
l 33: “health parameters” what health parameters are you think? Please clarify!
l 33 suggest delete “mainly”
l 37: “high pasture feeding” this sentence is not correct, please correct, e.g. high amount green forage feeding or grass!
l 39: add local breed name!
l 48-48: “health-benefiting molecules” what does it means? Please clarify!
l 68: p.232K and later: p.K232A, p.232AK, p232AA, pA238K, etc, please uniform the polymorphism!
l 96-97: please add correct data about feeding: ration, ingredients, pasture: main plant species.
l 141: please use milk composition instead of gross composition!
l 154 and Figure 1: no significant difference between SI AK and EX AA (indicated letters: a vs ab)!
Table 2: please add summa short chain fatty acids; n-3 fatty acids, n-6 fatty acids, n6/n3 ratio!
Table 2: please explained: CLA ratio!
l 197-199: please rewrite this sentence and may add into introduction section!
l 219: “due to breed” please explained this sentence!
l 228-230: no data forage composition in this manuscript!!!
l 238, 241: “C18:0 unsaturated FA” incorrect phrase!!!! C18:0 stearic acid!!
l 251: “effect of the DGAT1genotype on the health properties of milk…” but genotype not significantly affected the concentrations of oleic acid, vaccenic acid, CLA, etc, but affected the AI!!
l 293-294: “In fact, …” delete this sentence!
l 297: please add Modicana breed name!
l 297-299: “Furthermore, …” delete this sentence!
Reviewer 3 Report
The article describes the FA composition in discussion to system/feed and genetics within a rare breed.
The authors only focus on the DGAT1 polymorphism, which is just a single genetic trait. No other genetic differences are used to explain the differences. Other genetic traits are much more important in relation to the question of MUFA / SFA, like the activity of stearoyl-CoA desaturase.
The authors connect their results with the sn-position of the single FAs, whereas the article is mainly focusing on the FA composition.
The authors put much emphasis on the explanation of the small differences between the two genetic groups, which is mainly on the short chain saturated FAs, whereas the system differences are much more relevant, especially when it comes to FAs in relation to potential health: PUFA, CLA, ALA. The breed effect is over-emphasised.
There is no information about the real feed intake of the cows
There is no information about the weight and changes in weight of the cows.
This all might be plausible explanations between the genetic variants, but it is not measured.
There is too much emphasis on the health impact, which is not part of the research focus
I made several remarks to specific text parts:
L44: affect (not affects)
L48: skip everything about health benefits; this not the goal of your paper
L58-69. Please make clear how the positioning of the FA (sn-3) is related to increased fat percentage and higher saturated FAs. So, if it is only about positioning, why is the total milk FA changing?
Further: use one type of writing for K232A and p.232K? For instance only in capitals.
L74: skip healthier. You cannot proof this.
L88: “milk quality” is very wide, make this mor precise: for instance the milk fat composition.
L88-90: make this more precise: “modulation” goes in all direction and cannot be tested as an hypothesis.
L96-97: make this more precise. Try to calculate the intake of grass, hay and concentrates in the different systems. This is too vague. Weight of the cows??
L99-100 and L126. How do these data correspond? Eight months trial in relation to stage of lactation???
L131 and 132. Please describe this process in more detail. Not only show us two references, because the FA composition is an important part of your results.
L140. What do you mean with “pre-experimental data of milk collection”. What is the content?
L157 Table 1. What are these data? What is mean here? Is this the mean production of the cows during these eight month?
L215: “on the contrary…” should be “In contrats…”
L217: “milk traits” you mean “milk fatty acid composition”
L218-219. This is a nonsense sentence. Skip it
L222. According to your own table, no differences in casein percentage. Skip this
L222-233. Skip the whole discussion about protein and casein. You did not investigate this. Your focus is milk FAs.
L235-236.”This …fat composition” Skip this sentence. Nonsense sentence
L241: what are “lower C18:0 unsaturated FA”?
L254. This is your most relevant result: the system effect. The genetic effect are marginal compared to the system effect.
L284 Conclusion
Skip L285-290. This no conclusion, but discussion/introduction
L292-293. Please skip everything about health. Your results are too poor to claim more health and the genetic differences were very, very, very limited in comparison to the system differences. That should be the main part of your conclusion, not the focus on the A or K genetics.
L293-297. The whole issue about interaction is not been discussed well. It is only about some short chain FAs (C4:0 – C10:0) without any discussion on this.